# Epidemiology of Hand, Foot, and Mouth Disease and Genetic Characterization of Coxsackievirus A16 in Shenyang, Liaoning Province, China, 2013–2023

**DOI:** 10.3390/v16111666

**Published:** 2024-10-24

**Authors:** Fan Li, Qian Zhang, Jinbo Xiao, Huijie Chen, Shi Cong, Ling Chen, Huanhuan Lu, Shuangli Zhu, Tianjiao Ji, Qian Yang, Dongyan Wang, Dongmei Yan, Na Liu, Jichen Li, Yucai Liang, Lei Zhou, Mengyi Xiao, Yong Zhang, Baijun Sun

**Affiliations:** 1School of Public Health, Shenyang Medical College, Shenyang 110034, China; lifan1999082024@163.com (F.L.); 17702475505@163.com (S.C.); 15309856440@163.com (L.C.); 2National Polio Laboratory, World Health Organization Western Pacific Region Polio Reference Laboratory, National Health Commission Key Laboratory of Laboratory Biosafety, National Health Commission Key Laboratory of Medical Virology, National Institute for Viral Disease Control and Prevention, Chinese Center for Disease Control and Prevention, Beijing 102206, China; xiaojb@ivdc.chinacdc.cn (J.X.); luhuanhuan0908@163.com (H.L.); zhusli@126.com (S.Z.); jitj@ivdc.chinacdc.cn (T.J.); yangqian@ivdc.chinacdc.cn (Q.Y.); wangdy@ivdc.chinacdc.cn (D.W.); dongmeiyan1976@163.com (D.Y.); liuna@ivdc.chinacdc.cn (N.L.); jichenli666@163.com (J.L.); yucai.liang@hotmail.com (Y.L.); lei_zhou1999@163.com (L.Z.); m909184844@163.com (M.X.); 3China National Biotec Group Company Limited, Beijing 100029, China; zhangqian527@126.com; 4National Key Laboratory of Intelligent Tracking and Forecasting for Infectious Diseases, National Institute for Viral Disease Control and Prevention, Chinese Center for Disease Control and Prevention, Beijing 102206, China; 5Shenyang Center for Disease Control and Prevention, Shenyang 110000, China; chj1317@126.com; 6Department of Medical Microbiology, Shandong Second Medical University, Weifang 261053, China; 7School of Public Health, Shandong First Medical University (Shandong Academy of Medical Sciences), Jinan 250000, China

**Keywords:** hand, foot, and mouth disease, epidemiology, coxsackievirus A16, genetic characteristics, VP1

## Abstract

Hand, foot, and mouth disease (HFMD), a common childhood infection caused by enterovirus, poses a serious public health concern in China. We collected and analyzed epidemiological data on 62,133 HFMD cases in Shenyang City, Liaoning Province, from 2013 to 2023. The average annual incidence was 76.12 per 100,000 person-years; 99.45% of cases were mild, while 0.55% were severe. Only one patient died. HFMD infections peaked annually in July. Children in kindergartens and scattered children accounted for 44.6% and 42.2% of cases, respectively. Real-time RT-PCR detection of enteroviruses in 5534 patient samples revealed 3780 positives, of which 25.1% were CVA16-positive. Positives were randomly sampled, yielding 240 VP1 sequences of CVA16. Phylogenetic tree results showed that all VP1 sequences belonged to the B1 sub-genogroup. However, the sub-genogroup prevalence varied over time: from 2013 to 2014 and 2019 to 2021, the predominant sub-genogroup was B1a, while it was B1b from 2015 to 2018. Further phylogenetic analyses showed substantial divergence between B1a branches in CVA16, suggesting possible turnover of the B1a sub-genogroup in CVA16 due to evolution. This study provides epidemiological data on HFMD in Shenyang, and provides a phylogenetic analysis of CVA16, offering a theoretical basis for preventing and controlling HFMD in Shenyang City.

## 1. Introduction

Hand, foot, and mouth disease (HFMD), first reported in New Zealand in 1957 [1], is a common, acute infectious disease caused by various enteroviruses (EV) and is most common in children up to five years of age. The main clinical manifestations are fever accompanied by blisters or rashes on the hands, feet and mouth, and most patients have a short disease course with mild symptoms. However, some patients with severe disease may develop neurological and respiratory symptoms, such as neurogenic pulmonary edema, aseptic meningitis brainstem encephalitis, myocarditis, or other symptoms, which may even be fatal [2,3].

EV belongs to the genus *Enterovirus* in the family *Picornaviridae*. EV is a non-enveloped, single-stranded RNA virus with a total genome length of approximately 7400 bp. The genome encodes four structural proteins (VP1–VP4) and seven nonstructural proteins (2A–2C, 3A–3D), and the *VP1* region nucleotide sequence is commonly used for genotyping enteroviruses [4]. EVs infecting humans have been classified into four species: EV-A, EV-B, EV-C, and EV-D, based on differences in their genetic characteristics and biological properties. Most cases of HFMD are due to infection with EV-A, which includes two major genogroups, coxsackievirus A16 (CVA16) and enterovirus A71 (EV-A71) [5,6].

CVA16 is divided into three genogroups (A, B, and D), and the prototype strain (G10/RSA/1951) is genogroup A. Genogroup B is divided into sub-genogroups B1 and B2. B1 can be further subdivided into B1a, B1b, and B1c. Among these, sub-genogroup B2 was first reported in Japan in 1981, and has not been reported in the 21st century. Sub-genogroup B1 has been the most common sub-genogroup after 2000, and the evolutionary branches of B1, namely B1a and B1b, were first isolated in Japan in 1995 and 1998, respectively. It has dominated the CVA16 epidemic in the world for nearly two decades [6]. However, a new sub-genogroup, B1c, first appeared in Southeast Asia and parts of Europe after 2000 [7,8].

HFMD poses a serious burden to the global community. It has been reported worldwide, especially in East and Southeast Asian countries [9,10]. In China, where public health problems posed by HFMD have been serious since 2007, there have been more than 1 million new cases of HFMD annually [11,12]. Shenyang City is the central city of Liaoning Province, and is located at the center of the Northeast Asian Economic Circle and the Bohai Economic Circle. The characteristics of its HFMD epidemic are representative of the entire Northeast region in China. In this study, 5533 HFMD samples from Shenyang City, China, from 2013 to 2023, were tested and analyzed to clarify the epidemiological characteristics and pathogen spectrum composition of HFMD in Shenyang City over the past decade. EV-A71 and CVA16 have always been important pathogens of HFMD [13,14]. Since 2016, with the launch and application of the EV-A71 inactive vaccine, the prevalence of EV-A71 has shown a decreasing trend, and the pathogen spectrum of HFMD has undergone significant changes; CVA6 has gradually become the main pathogen of HFMD [15,16]. Since 2013, only one genogroup of EV-A71 and CVA6 has been prevalent in China. The C4a sub-genogroup of EV-A71 and the D3a sub-genogroup of CVA6 are the absolute dominant prevalent genogroups. However, unlike EV-A71 and CVA6, CVA16 has always had two sub-genogroups (B1a and B1b) prevalent in China, which piqued our curiosity [17,18]. Therefore, this study also focused on analyzing the evolutionary pattern of CVA16, filling the existing gap in the literature regarding the genetic characteristics of CVA16 in Shenyang City from 2013 to 2023.

## 2. Materials and Methods

### 2.1. Case Diagnosis

According to the HFMD Treatment Guidelines (2010 and 2018 editions), HFMD cases are reported to the HFMD surveillance system of the Chinese Center for Disease Control and Prevention. Mild cases of HFMD are characterized by markedly elevated temperatures accompanied by a rash or blister of the hands, feet, or mouth. In severe cases, cardiopulmonary and neurological involvement can lead to severe cardiopulmonary or neurological symptoms, usually including acute flaccid paralysis, aseptic meningitis, and pulmonary hemorrhage.

### 2.2. Epidemiological Analysis

Data on HFMD cases in Shenyang from January 2013 to December 2023 were obtained from the Infectious Disease Surveillance Information Reporting System, Chinese Center for Disease Control and Prevention. The total number of cases was 62,133. The population data of Shenyang City in this article comes from the statistical yearbook of Shenyang City (https://tjj.shenyang.gov.cn/sjfb/ndsj/, accessed on 24 November 2023). Data on reported cases, including sex, age, and time of onset, were categorized using Excel software (version 2409 Build 16.0.18025.20160). Statistical analyses were performed to characterize the epidemiology, including demographic characteristics, sex and age distributions, and seasonal variations. The overall incidence rate was defined as the total number of HFMD cases during the study period divided by the average population of Shenyang. Data were analyzed using SPSS statistical software (version 26.0). The chi-square test was used to assess the association between demographic variables. We set *p* < 0.05 as the cut-off for statistical significance.

### 2.3. Sample Collection and Viral Isolation

Between 2013 and 2023, 5533 samples were collected from patients with HFMD at the HFMD surveillance sentinel hospitals in Shenyang, Liaoning Province. The types of samples collected included feces, anal swabs, pharyngeal swabs (including both nasopharyngeal and oropharyngeal swabs), and blister fluid. Detection of enterovirus nucleic acid in the above specimens was performed using the enterovirus-specific real-time RT-PCR method with nucleic acid detection kits for enteroviruses (universal, CVA16, and EV-A71, JC20303N; BioPerfectus, Taizhou, China). Collected feces were processed according to standard procedures. CVA16-positive samples were inoculated into human rhabdomyosarcoma cells for culture, and after complete production of EV-like cytopathic effects, infected cell cultures were harvested for nucleic acid extraction [19].

### 2.4. Sequencing

Two hundred microliters of the obtained cell cultures were subjected to extraction using a GeneRotex 96 Nucleic Acid Extractor (Tianlong, Xi’an, China), and the extracted viral RNA was stored in a freezer at −80 °C. Subsequently, we used Prime Script One Step RT-PCR Kit Ver.2 (TaKaRa, Dalian, China) and primers 486/489 to amplify the entire VP1-coding region [20]. The PCR products were purified using the QIAquick PCR Purification Kit (Qiagen, Hilden, Germany), and an ABI 3130 Genetic Analyzer (Applied Biosystems, HITACHI, Tokyo, Japan) was used for sequencing [21].

### 2.5. Phylogenetic and Phylodynamic Analyses

We performed a simple random sampling of all samples tested for CVA16 in Shenyang City from 2013 to 2023, taking 30 samples per year, to finally obtain 240 samples for CVA16 evolutionary characterization. Fifty-two CVA16 molecular typing reference sequences were downloaded from the GenBank database, and the genogroups of 240 CVA16 sequences were identified.

Sequences were aligned using Muscle software in MEGA (version 11.0) [22]. The identity matrix of nucleotide sequences was generated using BioEdit (version 7.0.9.0) [23]. Maximum likelihood (ML) trees were constructed in MEGA (version 7.0) with 1000 bootstrap replications using a general time-reversible model with a gamma distribution and invariant sites (GTR + G + I), as suggested by jModelTest (version 2.1.10) [24]. The temporal structure of the sequences was examined using TempEst (version 1.5) [25] to ensure that the sequences involved in the construction of the ML tree and their sampling times were correct. Mean distances within and between groups for 1000 bootstrap replicates were calculated using the p-distance model in MEGA (version 11.0). The online tool iTOL website (https://itol.embl.de/, accessed on 11 October 2024) was used for post-annotation of the constructed evolutionary tree.

Phylodynamic analyses were performed using BEAST software (v2.5) [26], and the Shenyang CVA16 dataset was constructed by selecting four samples per month per year, and the molecular evolution rate of Shenyang CVA16 was estimated using the uncorrelated relaxation clock model and Bayesian skyline. Tracer software (version 1.7.1) [27] was used to observe the degree of convergence of the parameters when the effective sample size reached 200 or more. The parameters were then considered to be converged. The maximum clade credibility (MCC) tree was then constructed using Tree Annotator (version 1.10.4) and visualized using Figtree software (version 1.44) (http://tree.bio.ed.ac.uk/software/Figtree/, accessed on 11 October 2024).

## 3. Results

### 3.1. Epidemiological Characteristics of HFMD in Shenyang City

From 1 January 2013 to 31 December 2023, a total of 62,133 cases of HFMD were reported in Shenyang. Among them, 61,785 cases (99.4%) were mild, and 348 cases (0.05%) were severe. There was only one fatal case, which occurred in 2013. The average annual incidence rate was 76.12/100,000 (range: 47.25–107.98/100,000). In terms of seasonal distribution, the incidence rate peaked in the summer months of June and July. In other seasons, the incidence remained low (Figure 1A).

Since 2013, the overall incidence rate of HFMD in Shenyang has fluctuated. From 2014 to 2017, the incidence rate declined year by year. In 2017, the incidence rate reached the lowest value in the past decade; however, in 2018, the incidence rate began to rise to the level of 2015. However, due to the COVID-19 pandemic, the incidence rate declined after 2019, and in 2023, the incidence rate reached its highest level since 2013. (Figure 1B).

During this period, the rate of serious HFMD cases in Shenyang City remained at a low level (mean value of 0.58%). The proportion of serious cases exceeded the mean value in the three years from 2018 to 2020, in which the rates were 1.99%, 0.63%, and 1.32%, respectively (Figure 1C).

After analyzing the reported data of HFMD in Shenyang City, we found the average number of cases between 2013 and 2023 was 4767 cases per district. The distribution pattern of cases in Shenyang City was higher in the center and lower in the periphery. Among the districts studied, the Tiexi and Yuhong districts had the highest number of cases, which was 2–3 times the average number of cases, while Liaodong and Faku counties were only 1/10 of the average number of cases (Figure 2).

In our data, 36,655 patients were male, while 25,478 were female, meaning that the male-to-female ratio was 1.44:1 (Figure 3A). From 2013 to 2023, patients with HFMD in Shenyang City were mainly children aged 1–5 years old, accounting for 71.7% (42,329/62,133) of cases. The total number of patients aged under 1 year old was 3187, accounting for only 11.4% of the total number of patients aged under 5 years old. Stratified by years of age, children aged 1–2 years made up the highest proportion of patients at 23,095 (38.64%). Children aged 3–4 years were next at 22,421 (33.11%), followed by children aged 5–6 years at 8878 (10.24%), and finally, children aged 6 years or above at 7739 (6.61%) (Table 1).

Among all HFMD patients, the distribution by category was as follows: scattered children (44%), children attending kindergarten or nursery school (42%), and the remaining children who were attending school or belonged to another category made up 14% of the total (Figure 3B).

### 3.2. HFMD Pathogen Spectrum of Shenyang

A total of 5531 HFMD samples were received by Shenyang CDC between 2013 and 2023, including blister fluid, anal swabs, and fecal samples. A total of 3779 positive specimens were detected out of 5531 samples, indicating a positive rate of 68.32%. Among the positive samples, the type with the highest detection rate was CVA6 (1346), accounting for 35.62%; the second highest was CVA16 (951), accounting for 25.17%; while the third was EV-A71 (693), accounting for 18.34% (Figure 4A). In addition to the common serotypes in China, some relatively rare serotypes were also detected (e.g., CVB3, CVB2, CVA8, CVA5, CVA2, CVA12, CVA4, CVA10, CVA9, CVB5, CVB6, E9, etc.), which accounted for 20.88% of the positive specimens (Figure 4B, Table 2).

Among the 348 severe HFMD cases reported in Shenyang from 2013 to 2023, clinical specimens were collected from 268 cases, of which 232 were positive for enterovirus, with a positivity rate of 86.57%. In 2013, 2015, and 2017, EV-A71 was the most common pathogen in severe HFMD cases; CVA16 was the most common pathogen in severe HFMD cases in 2014 and 2016; and CVA6 was the most common pathogen in severe HFMD cases in 2018 and 2019. After 2020, the number of severe HFMD cases decreased significantly (Figure 4C). In addition, CVA2, CVA4, CVA8, CVA10, and CVB2 have all been sporadically detected from severe HFMD cases (Figure 4D).

### 3.3. Phylogenetic and Phylodynamic Analysis of CVA16

According to the phylogenetic tree constructed by combining the sequences obtained from Shenyang City with those downloaded from GenBank, 240 CVA16 sequences were categorized into three branches, all of which belonged to sub-genogroup B1. When they were further sub-genogrouped, we found that the distribution of sub-genogroups B1a and B1b varied significantly across years, with sub-genogroup B1a mainly appearing in 2013, 2014, and 2019, as well as 2021 when it was significantly detected. In contrast, the annual distribution of sub-genogroup B1b was more extensive, beginning in 2013 and continuing through 2022, with significantly increased detection rates of sub-genogroup B1b in 2015 and 2018. In addition, we noted that sub-genogroup B1c was extremely rare, with only three cases found in 2017 and 2022 (Figure 5A).

From the MCC tree, it can be seen that sub-genogroups B1a and B1b of CVA16 in Shenyang City first diverged around 1965. Among them, B1a was divided into two different branches in 1980; one included mostly sequences from 2013 sequences, while the other included mostly sequences from 2019. B1b, in contrast, was divided into two branches in 1990, one of which showed multiple differentiated distributions between 2013 and 2018, and the other led to B1b after 2019 (Figure 5B). By comparing the distributions of sub-genogroups B1a and B1b across years, we can observe a clear trend of turnover. Sub-genogroup B1a seems to have been gradually replaced by sub-genogroup B1b after 2013 and 2014, especially between 2015 and 2018. Furthermore, B1c was only detected in 2017 and 2022.

Subsequently, we analyzed the genetic distances of the sequences in different years, and the genetic distance matrix showed that the intergroup distances of CVA16 sequences in Shenyang City increased year-to-year (Figure 5C). In addition, we analyzed the genetic distances of B1a sequences, finding that the intragroup distance of sequences that segregated from 2013 to 2014 was 0.0580. In contrast, the intragroup distance of sequences that segregated from 2019 to 2021 was 0.0234, whereas the intergroup distances of sequences that segregated before 2014 and after 2019 were 0.0777, which exceeds their respective intragroup distances. These findings are consistent with the results showing that two branches of the MCC tree had formed.

## 4. Discussion

In this study, we comprehensively analyzed the epidemiological characteristics of HFMD in Shenyang City based on the HFMD surveillance data of Shenyang City in the past 11 years. The results showed that patients with HFMD in Shenyang City mainly had mild cases, accounting for 99.4%. The age distribution of HFMD patients was dominated by children aged 1–5 years, accounting for >70% of the total number of patients. Therefore, children in this age group should receive more public health attention. The infection rate of boys was 1.44 times that of girls. It is hypothesized that this may be due to differences in susceptibility to host immune status between boys and girls, leading to differences in infection rates between the sexes [28].

HFMD cases are reported throughout the year in Shenyang City, but the overall distribution shows a single peak in the summer, with the largest number of patients reported in July each year, with even distribution in all other seasons. This phenomenon is similar to that reported in other cities [29]. However, in the last few years, the peak of infection has occurred in August, a month later than in previous years. This phenomenon has also been noted in other cities in China [30]. This seasonal variation may be correlated with factors such as climate, hygiene conditions, or viral mutations [31,32,33,34]. The year 2019, before the COVID-19 pandemic, was the first time there was a shift towards a peak in the epidemic. In fact, this peak can only be truly seen in two years: 2019 and 2023. During the COVID-19 pandemic, public health measures such as wearing masks, washing hands frequently, and maintaining social distance were implemented across the country to prevent and control the spread of COVID-19. Schools reduced in-person classes, public activities decreased, and travel restrictions were implemented, reducing opportunities for children and adults to gather together. These measures not only prevented and controlled the spread of COVID-19, but also effectively reduced the spread of infectious diseases such as hand, foot, and mouth disease that are transmitted through close contact. However, with the effective control of the COVID-19 pandemic, previously implemented strict public health measures have gradually been relaxed. In addition, during the COVID-19 pandemic, the spread of HFMD was restricted due to social isolation and reduced population contact, which may have led to a decrease in the immune level of the population against HFMD-related enteroviruses. After the COVID-19 pandemic, the proportion of susceptible populations may increase. All of these may lead to an increase in the incidence of HFMD, resulting in a new peak of incidence in 2023.

In addition, we also studied the geographic distribution within Shenyang City. Over the past decade, the highest total number of HFMD infections has been recorded in the Tiexi and Yuhong districts. These two districts together accounted for 40% of all infections, and the number of infections in the downtown area was four to five times higher than in the peripheral districts, so the closer to the downtown area, the more infections there were.

In the last decade, HFMD has been recognized as being primarily due to EV-A71, CVA16, CVA6, and CVA10 in China [13,14,15]. The analysis of the pathogen’s spectrum of HFMD in Shenyang City shows that CVA16 has been co-prevalent with other enteroviruses in Shenyang City since 2013, accounting for an average of over 10% of the pathogen’s spectrum of HFMD each year. In recent years, this proportion has exceeded 15%. This indicates that CVA16 is one of the main pathogens of HFMD in Shenyang. CVA16 is also the main pathogen causing severe HFMD cases in Shenyang, especially in 2014 and 2016, when it was the first pathogen to cause severe HFMD cases in Shenyang. Therefore, phylogenetic analysis of CVA16 strains in Shenyang City can help us better understand the characteristics of this pathogen and enrich our knowledge. Here, we used the sequences of strains with identified genogroups on GenBank as reference strains to further delineate the specimens identified as CVA16 in Shenyang City, which can be categorized into two sub-genogroups: B1a and B1b.

In the Chinese Mainland, most reports of CVA16 have been associated with two evolutionary branches of B1: B1a and B1b [6,18,35]. All strains isolated in this study belonged to genogroup B1. We identified 118 samples as belonging to sub-genogroup B1a, occurring mainly in 2013, 2014, 2019, and 2021. In contrast, 119 samples were identified as belonging to sub-genogroup B1b, occurring mainly in 2013, 2015, 2016, 2017, 2018, 2019, and 2022. B1a appeared for the first time in Shenyang in 2013; this sub-genogroup, however, declined dramatically after 2014 until it disappeared, while B1a appeared again in Shenyang and became prevalent in 2019, showing a cycle of appearance–disappearance–reappearance. The MCC tree shows that the two prevalent B1a sequences form two distinct branches, and the intergroup distance between the two prevalent B1a sequences is substantial. The re-emergence of B1a sub-genogroups is a noteworthy phenomenon, which is caused by the natural process of viral mutation and evolution. The occurrence of this phenomenon may be related to immune evasion, which may mean that the virus has undergone changes, reducing the immune protection previously obtained through natural infection. The re-emergence of B1a sub-genogroups may be more easily transmitted or have increased pathogenicity, which may lead to an increase in the number of cases or the resurgence of the epidemic, thus bringing new prevention and control challenges [36,37,38]. The increased diversity of newly prevalent B1a sequences from the previous ones suggests that CVA16 is still evolving, and the emergence of B1a may pose new public health problems, necessitating the need to strengthen the surveillance of CVA16. Unlike sub-genogroup B1a, sub-genogroup B1b appeared in all years, suggesting that B1b has spread in different years. The MCC tree showed that B1b can be divided into two branches, one of which mainly contains some strains in seen 2019, while the other contains most sub-genogroup B1b strains. CVA16 B1b was the main sub-genogroup prevalent in the Chinese Mainland between 2012 and 2019 [17].

This study summarized and analyzed the epidemiological information of HFMD in Shenyang City from 2013 to 2023, describing and analyzing factors such as incidence rate, populations, time of onset, and geographic distribution. Phylogenetic analyses and evaluations of the evolutionary dynamics of the VP1 coding region of Shenyang CVA16 were performed to determine the evolutionary and epidemiological characteristics of CVA16 strains in Shenyang. The study revealed the transformation of sub-genogroups of CVA16 over different years, enriching our understanding of HFMD in Northeast China and providing data to support the further development of public health prevention strategies and vaccine research.

There are some limitations in this study. Firstly, the data in this study may differ from the actual situation because HFMD symptoms are mild and patients may not seek medical attention after infection, or because the diagnosis of HFMD in hospitals depends largely on the personal factors of doctors, and patients with mild HFMD may be misdiagnosed, e.g., the younger the patient, the more attention is given to the patient, which ultimately leads to a bias in case selection. Secondly, the data in this study were only for Shenyang city, where climatic factors and hygienic conditions may affect the incidence of HFMD. These issues need more attention in subsequent studies.

## Figures and Tables

**Figure 1 viruses-16-01666-f001:**
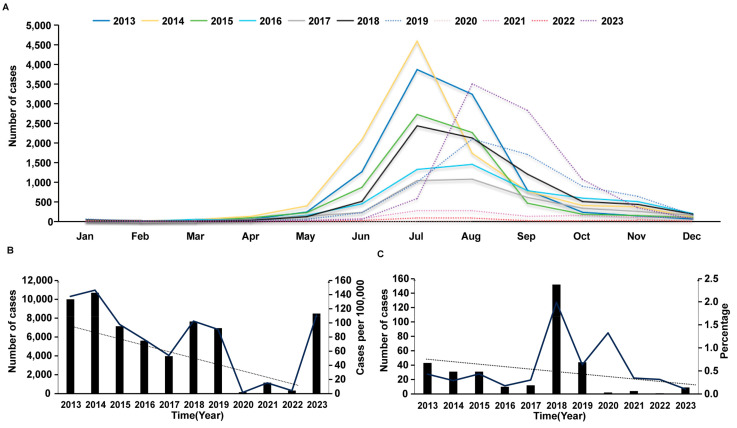
Time distribution of hand, foot, and mouth disease (HFMD) incidence in Shenyang from 2013 to 2023. (**A**) Monthly distribution of patients with HFMD in Shenyang between 2013 and 2023. (**B**) Trends in the annual distribution of patients with HFMD in Shenyang between 2013 and 2023; the dotted line represents the trend line, the solid line represents the incidence rate, and the bar chart is the number of HFMD cases. (**C**) Yearly distribution of severe HFMD cases in Shenyang between 2013 and 2023; the dotted line represents the trend line, the solid line represents the incidence rate, and the bar chart represents the number of severe HFMD cases.

**Figure 2 viruses-16-01666-f002:**
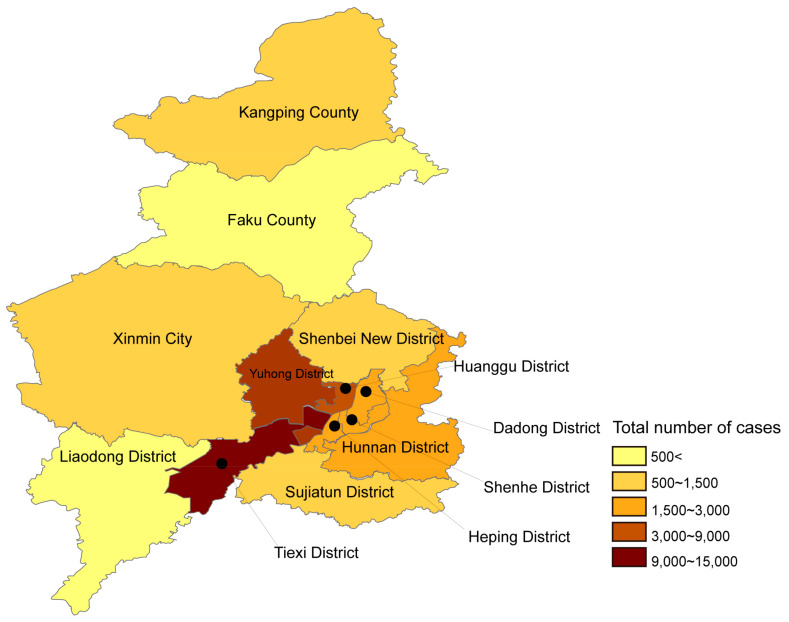
HFMD epidemiological regions and distribution in Shenyang, 2013–2023. Shades of color are used to indicate the number of cases.

**Figure 3 viruses-16-01666-f003:**
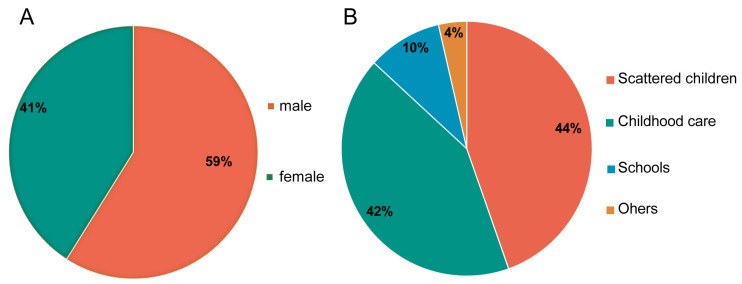
Population distribution of HMFD in Shenyang between 2013 and 2023. (**A**) Sex distribution of patients with HFMD in Shenyang City from 2013 to 2023. (**B**) Categories of patients with HFMD in Shenyang between 2013 and 2023.

**Figure 4 viruses-16-01666-f004:**
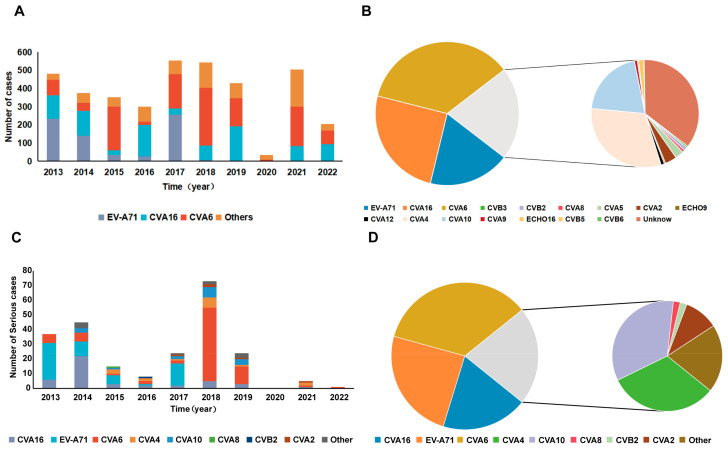
Types of enteroviruses detected in hand, foot, and mouth disease patients in Shenyang, China from 2013 to 2023. (**A**) Types of enteroviruses with a high detection rate. (**B**) All detected types of enteroviruses. (**C**) Types of enteroviruses from severe HFMD cases. (**D**) All detected types of enteroviruses from severe HFMD cases.

**Figure 5 viruses-16-01666-f005:**
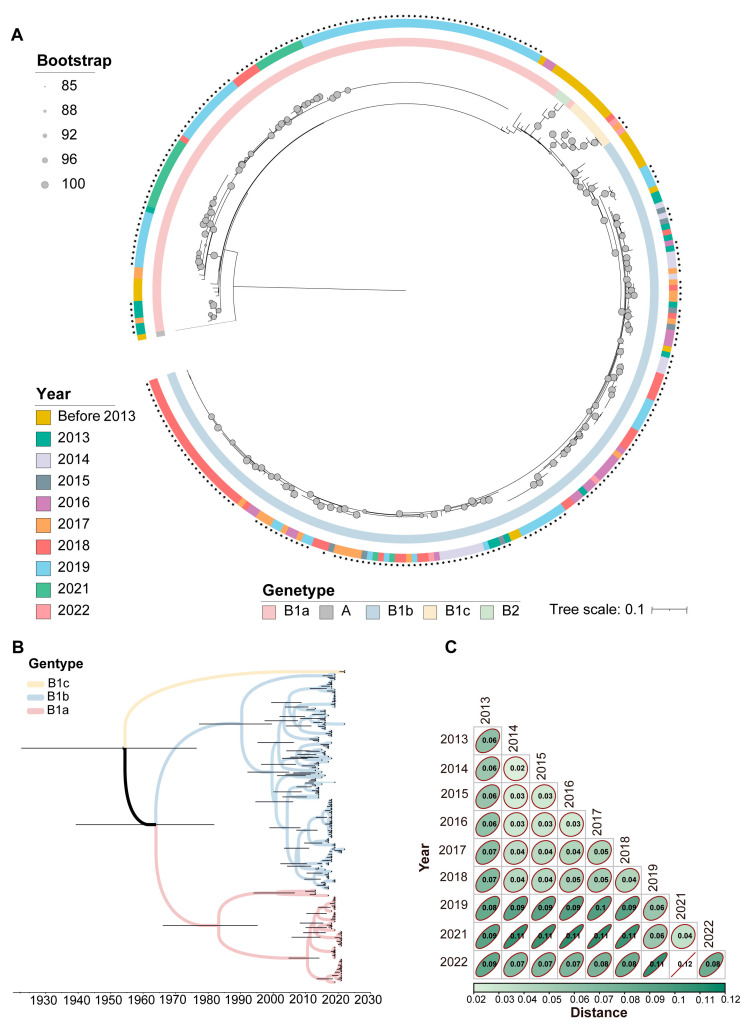
(**A**) Maximum likelihood tree constructed based on 240 CVA16 VP1 sequences in Shenyang from 2013 to 2023. Bootstrap values are indicated by solid circles, and the circle size represents the size of the bootstrap value. The outer circle represents CVA16 from different years, and CVA16 sequences from Shenyang are marked with an asterisks. The inner circle represents different genogroups/sub-genogroups marked with different colors. (**B**) Time-scaled phylogenetic tree based on complete VP1 nucleotide sequences of CVA16 strains in Shenyang. (**C**) Genetic distance matrix of CVA16 nucleotide sequences from different years.

**Table 1 viruses-16-01666-t001:** Summary of year of onset, sex, and age of HFMD patients in Shenyang from 2012 to 2023.

Variable											
	HFMD Reported Cases				Sex			Age Group (%)			
	Mild Cases	Severe Cases	Deaths	Total (Incidence Rate/10^5^)	Male	Female	Sex Ratio	<1	1~2	3~4	5~6
2013	9955	43	0	9998 (137.51)	5840	4158	1.40	1058 (1.82%)	3805 (3.09%)	3414 (2.78%)	1059 (0.86%)
2014	10,665	31	1	10,696 (146.36)	6062	4634	1.31	289 (0.40%)	3453 (2.26%)	4459 (2.92%)	1669 (1.09%)
2015	7114	31	0	7145 (97.82)	4283	2862	1.50	461 (0.94%)	2902 (2.80%)	2550 (2.46%)	768 (0.74%)
2016	5607	10	0	5617 (76.48)	3305	2312	1.43	187 (0.27%)	1865 (1.27%)	2483 (1.69%)	655 (0.45%)
2017	3962	12	0	3974 (53.92)	2270	1704	1.33	178 (0.28%)	1202 (0.88%)	1657 (1.21%)	567 (0.41%)
2018	7494	152	0	7646 (102.49)	4648	2998	1.55	396 (0.67%)	2714 (2.16%)	2727 (2.17%)	1018 (0.81%)
2019	6898	44	0	6942 (90.70)	4175	2767	1.51	317 (0.51%)	2074 (1.56%)	2459 (1.85%)	1091 (0.82%)
2020	149	2	0	151 (1.66)	89	62	1.44	9 (0.02%)	61 (0.05%)	40 (0.03%)	17 (0.01%)
2021	1156	4	0	1160 (15.16)	705	512	1.38	35 (0.08%)	267 (0.27%)	376 (0.38%)	250 (0.26%)
2022	316	1	0	317 (4.15)	178	139	1.28	15 (0.04%)	81 (0.10%)	100 (0.12%)	66 (0.08%)
2023	8478	9	0	8487 (111.09)	5142	3345	1.54	242 (0.57%)	1484 (1.75%)	2156 (2.55%)	1718 (2.03%)
summary	61,794	339	1	62,133 (837.4)	36,697	25,493	/	3187 (5.60%)	19,908 (16.19%)	22,421 (18.16%)	8878 (7.56%)

**Table 2 viruses-16-01666-t002:** The number of various enterovirus types detected in hand, foot, and mouth disease patients in Shenyang from 2013 to 2022.

	2013	2014	2015	2016	2017	2018	2019	2020	2021	2022
EV-A71	233	139	33	27	256	5	0	0	0	0
CVA16	130	138	26	173	33	82	192	0	84	93
CVA6	85	44	240	17	189	316	155	9	215	76
CVB3	0	0	1	2	1	0	0	0	0	0
CVB2	0	1	2	2	3	0	0	0	0	0
CVA8	0	1	2	0	3	0	0	0	0	0
CVA5	0	1	1	3	3	1	11	0	0	0
CVA2	1	0	4	4	5	4	5	0	5	0
E9	1	0	1	1	0	0	0	0	0	0
CVA12	0	1	2	3	2	0	0	0	0	0
CVA4	10	5	16	11	14	35	5	0	147	4
CVA10	11	30	8	26	15	38	23	0	9	0
CVA9	1	1	0	0	3	0	2	0	0	0
E16	0	0	0	2	1	0	0	0	0	0
CVB5	0	0	0	0	0	12	0	0	0	0
CVB6	0	0	0	0	0	0	1	0	0	0
Unknown	10	14	15	30	26	50	35	25	45	32
Summary	482	375	351	301	554	543	429	34	505	205

## Data Availability

Data is deposited in the National Microbiology Data Center (NMDC) with accession numbers NMDCN00060GI to NMDCN00060O1 (https://nmdc.cn/resource/en/genomics/sequence/detail/NMDCN00060GI to https://nmdc.cn/resource/en/genomics/sequence/detail/NMDCN00060O1, accessed on 11 October 2024).

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
