# Peer review of "Epidemiology of Hand, Foot, and Mouth Disease and Genetic Characterization of Coxsackievirus A16 in Shenyang, Liaoning Province, China, 2013–2023"

_viruses, 2024, doi:10.3390/v16111666_

Round 1

Reviewer 1 Report

Comments and Suggestions for Authors

The manuscript analyzed 10 years of epidemiological data on HFMD in Shenyang, reporting in particular on CVA16, and found possible changes in the prevalence between b1A and b1B. As serotype CVA16, it is further divided into ABD, and even B1(B1a,B1b)this study can provide value for the prevention and control of enteroviruses. 

1. The overall structure is clear, but the logical connection between some paragraphs can be further strengthened.

2. The article mentions that the peak of HFMD incidence in recent years has been delayed from July to August, but does not delve into possible causes. It is recommended to explore or cite relevant studies to explain whether this seasonal change is related to factors such as climate, sanitation conditions or virus variation.

3. The authors provided a more detailed analysis of the prevalence of the CVA16 subgenotypes B1a and B1b, but "B1a reappeared." "B1b persists." The public health significance of the phenomenon can be further explored. In particular, whether the re-emergence of B1a means that the subgenotype is different in resistance to vaccines, or whether its evolution rate presents new prevention and control challenges, it is recommended that the authors add more discussion to this section.

4. The authors' hypothesis that the infection rate is higher in boys is based on speculation about activity and personal hygiene habits, and recommend that additional literature be added to support such assumptions. The discussion mentioned "boys are more active" Or "poor hygiene habits" The reason needs to be supported by more evidence or survey data to avoid appearing too subjective. 

Comments on the Quality of English Language

ok

Author Response

Reviewer 1 comment:

The manuscript analyzed 10 years of epidemiological data on HFMD in Shenyang, reporting in particular on CVA16, and found possible changes in the prevalence between b1A and b1B. As serotype CVA16, it is further divided into ABD, and even B1 (B1a,B1b) this study can provide value for the prevention and control of enteroviruses. 

  1. The overall structure is clear, but the logical connection between some paragraphs can be further strengthened.

------Response to reviewer 1 comment No. 1: Thank you for your valuable comments, modifications have been made and the logical connection between some paragraphs in the ABSTRACT and DISCUSSION were strengthened.

  1. The article mentions that the peak of HFMD incidence in recent years has been delayed from July to August, but does not delve into possible causes. It is recommended to explore or cite relevant studies to explain whether this seasonal change is related to factors such as climate, sanitation conditions or virus variation.

------Response to reviewer 1 comment No. 2: Thank you for your valuable comments, and we cite relevant research to explain the correlation between this seasonal variation and factors such as climate, hygiene conditions, or viral mutations. (Lines 292-293; References 31-34).

  1. The authors provided a more detailed analysis of the prevalence of the CVA16 subgenotypes B1a and B1b, but "B1a reappeared." "B1b persists." The public health significance of the phenomenon can be further explored. In particular, whether the re-emergence of B1a means that the subgenotype is different in resistance to vaccines, or whether its evolution rate presents new prevention and control challenges, it is recommended that the authors add more discussion to this section.

------Response to reviewer 1 comment No. 3: Thank you for your valuable comments, and more discussion to this section was added.

The re-emergence of B1a sub-genotypes is a noteworthy phenomenon, which is caused by the natural process of viral mutation and evolution. The occurrence of this phenomenon may be related to immune evasion, which may mean that the virus has undergone changes, reducing the immune protection previously obtained through natural infection. The re-emergence of B1a sub-genotypes may be more easily transmitted or have increased pathogenicity, which may lead to an increase in the number of cases or the resurgence of the epidemic, thus bringing new prevention and control challenges. (Lines 339-345; References 36-38).

  1. The authors' hypothesis that the infection rate is higher in boys is based on speculation about activity and personal hygiene habits, and recommend that additional literature be added to support such assumptions. The discussion mentioned "boys are more active" Or "poor hygiene habits" The reason needs to be supported by more evidence or survey data to avoid appearing too subjective. 

------Response to reviewer 1 comment No. 4: Thank you for your valuable comments and the relevant content without data and reference support has been deleted. The infection rate of boys is 1.44 times that of girls. It is hypothesized that this may be due to differences in susceptibility to host immune status between boys and girls, leading to differences in infection rates between the sexes. (Lines 283-286; Reference 28).

Reviewer 2 Report

Comments and Suggestions for Authors

The manuscript deals with the epidemiology of hand, food and mouth disease (HFMD) as well as the genetic characterization of Coxsackievirus A16 (CVA16) in a Chinese city, Shenyang, from years 2013 to 2023. This paper has a lot of strengths such as a lot of data and spans over a decade. Also, it nicely shows how the pandemic has effected the circulation of HFMD causing enteroviruses. It thorough and well done. However it would benefit of some additions and revisions. The English is in my opinion very good and I usually leave the language corrections to native speakers. However, this time I have suggested a few revisions to the language. Altogether, this study adds valuable data to the epidemiological features of HFMD, the enteroviruses that are causing it and changes during the last 11 years.

The epidemiological part would be stronger if the serious disease causing virus types would be mentioned, if this data is available to the writers. For example there are two peaks where the ratio of serious cases is higher compared to other years, 2018 and 2020. 2020 is possibly a coincidence, since the numbers are so small. However, year 2018 is more interesting. In this year incidence was high and the most common cause was CVA6, even compared to other years. Moreover, in that year EV-A71 seems to be very low or completely missing. EV-A71 is known to be a highly pathogenic EV type, causing neurological manifestations which sometimes leads to death.

Also the question why was CVA16 selected for the genetic characterization is not well addressed. It was not the most common type, CVA6 was. Did this have something to do with the pathogenicity? Nothing of this sort was mentioned.

The writhers should have payed attention to some details. For example the taxonomical levels are wrong for some parts. The levels of taxonomy are: family: Picornaviridae, genus: Enterovirus, species: Enterovirus A, type or genotype (formerly serotype): Coxsackievirus A16 (and others), genogroup or clade: CVA16 B1, subgenogroup: CVA16 B1a.

Detailed comments to the manuscript are presented below:

Introduction:

line 45: “various species of Enterovirus (EV)” could it be that here the writers mean type of Enteroviruses, CVA6, CVA16, EV-A71 all belong to species Enterovirus A.

line 47: ”mouth herpes”, is strictly speaking caused by the herpes virus. Herpangina or mouth blister would be a better option to be used here.

Lines 52-55: This is a very long sentence, consider revising to two or more sentences.

lines 56-57:  “four groups, EV-A, EV-B, EV-C, and EV-D” these are species.

beginning from line 60: coxsackievirus A16 and enterovirus-A71 are virus types, formerly called serotypes. These can also be called genotypes since the typing is done by sequencing the VP1 like the writers say. Furthermore, the CAV16 can be divided into (at least) three genogroups or clades (subtypes). These again can be divided into subgenogroups or subclades.

Line 61-62: “B1a can be further subdivided into B1a, B1b, and B1c.” The sentence begins with “B1a” – probably should be just “B1”.

Materials &methods:

Line 85: herpes is a virus, if by herpes the writers mean a symptom/disease, herpangina or blisters, please change. Herpes can possibly be used to describe a symptom such as blisters, but in scientific text it should not be used.

Line 105: The sample type pharyngeal swabs, does it include both nasopharyngeal and oropharyngeal swabs? Again, not herpes, rather blisters.

Line 106-107: There is no need to spell out PCR, and furthermore, the test for EV:s is RT-PCR (traditional, real-time, quantitative or not).

Line 114: It should be said what was extracted form cell culture media. RNA, DNA, nucleic acid or protein, there are more than one option.

Line 116: It is not important in which tubes the RNA was stored, but at what temperature? -20°C or -70/-80°C.

Results:

The text (lines 156-159) and figure 1 do not correspond to each other. Years 2020-2022 were very low years for HFMD, the increasing trend did not increase throughout 2018 to 2023. This is totally understandable, since it was the time of the COVID-19 pandemic.

Figure 1 and most other figures: Figure legend would need a little more details. Also, it should describe what is actually presented in the figures (numbers, percentages etc.). Figure legends with more than one figure in them should first have an overall title. Then after that the description of each subfigure beginning from (A). I have written an example for this in figure 4 below.

Figure 1: Especially in B and C there is no explanation of what the different bars and lines are. Also, interesting would be in figures B and C, which proportion of cases were detected EV positive.  

Figure 2: to my eye it looks like there are 7 different shades of yellow and brown on the map, but only 5 have the numerical explanation of cases (in figure).  

Line 180: the male-female ratio is in figure 3A, not 3B.

Lines 182-183: Why not tell the percentage for the less than 1 year old children of the whole group, like for other age groups. That would make more sense for me.

Figure 3B and in some parts of the text: I don’t know the term “scattered children”, and in the figure named “diaspora” which I possibly understand? Also, there is nowhere that these mean the same, I only concluded this from the numbers that these probably mean the same. If these are the same, please use just one word for this group of children. Maybe this should be better explained, how the scattered children or diaspora children were defined.

Table 1: A few points on this table. First, I don’t see the point in the table to present the yearly numbers, the data is already presented in Figure 1. The only striking thing is the high number (and percentage) of severe cases in 2018, but this is also in Figure 1. Other numbers are quite similar and can be discussed in the text and figures. If the writers insist of keeping this table as it is, a summary of all the data should be there. Just add one row to the end of the table with “all together” etc. and add up the numbers. I would move this table to supplementary material or remove it.

Table 1 (continued): What has to be corrected or explained are the percentages below the age groups, from what is this the percentage calculated from? For example, year 2020 with very low numbers, percentages of children below 6 years added together is only 0,11%. This is at least not the percentage of the total in year 2020, which is the one that would make sense.

Line 204: Again, instead of herpes use for example blister fluid or something else.

Figure 4: Be sure the figure legend will be on the same page as the figure itself. The figure legend could be revised, for example, in the following way: First have a title of the picture “The types of enteroviruses detected among HFMD patients in Shenyang, China in years 2013-2023”. Then have the A and B, for example like this “(A) Enterovirus types detected at high rates, (B) less frequently detected types”. I would not use the word pathogens, because this paper deals with enteroviruses in HFMD.

Figure 4A: why is year 2023 missing? In the discussion is said that the proportion of CVA16 cases has a rising trend, this cannot be seen anywhere. This figure could show some of it, but as I suggested adding a table with the types found might be helpful (possibly as a supplementary).

Furthermore, the figure B has all types detected, not only the less commonly detected ones. Also, if I interpret the figures correctly, the next common types are CVA4 and CVA10, which make up about half of the others. That would mean CVA4 would cause about 6-7% of all HFMD enterovirus cases, which is a good proportion of cases, also CVA10 might be 4%?. Should these be in the more common ones? The numbers of virus types found could possibly be presented in a table, which could be as a supplementary. This might interest some of the readers. Possibly also the yearly numbers of virus types in the same table (as a supplementary). 

Figure 5: Details of pictures are very hard to distinguish from each other. Bootstrap values, from the picture it is hard to tell the difference in circle size. Are these circles necessary? The year and genotype colours are too much alike. It is impossible to distinguish several years from each other, for example 2014, 2015 and 2016 as well as 2017 and 2018. It is not even described in figure legend that year colours are on the outer circle and genogroups and sub-genogroups are the inner coloured circle (here again the levels of taxonomy, strain is enterovirus A). These should be used right.

Discussion:

These could have been a few sentences about the low Covid-19 pandemic years and the possible shift in the peak of infections. From figure 1 it looks like the shift was seen the first time in 2019, which is pre pandemic. Actually, this peak can be really seen in only two years 2019 and 2023, as seen in fig 1. That cannot really be seen as a trend, only individual years. The peak shift has been mentioned, but nothing on why that might have been.

Line 259-260: I very much doubt that girls in the ages below 5-6 years would be aware of personal hygiene. At least the European girls are not yet aware at that age, maybe Chinese children are different. It has also been found that boys tend to have slightly more EV and other infections, which is possibly mostly due to immunity.

Lines 278-280: The rising trend of CVA16 causing HFMD is not seen anywhere in the data, this is only suggested in the discussion. A table would help with different types causing HFMD would help, and if the rising trend is seen there it could also be in the manuscript itself.

Beginning on line 296: Speculation that changes in dominant genogroup and subgenogroup to changes in morbidity are interesting. However, it would be helpful to know which virus type, possible genogroup or subgenogroup have caused the severe cases. If there would be a correlation then it would be more informative and convincing.

 Line 306: The word summary is mentioned in this sentence twice. Just take away “In summary” from the beginning of the sentence.

Author Response

Reviewer 2 comment:

The manuscript deals with the epidemiology of hand, food and mouth disease (HFMD) as well as the genetic characterization of Coxsackievirus A16 (CVA16) in a Chinese city, Shenyang, from years 2013 to 2023. This paper has a lot of strengths such as a lot of data and spans over a decade. Also, it nicely shows how the pandemic has effected the circulation of HFMD causing enteroviruses. It thorough and well done. However it would benefit of some additions and revisions. The English is in my opinion very good and I usually leave the language corrections to native speakers. However, this time I have suggested a few revisions to the language. Altogether, this study adds valuable data to the epidemiological features of HFMD, the enteroviruses that are causing it and changes during the last 11 years.

------Response to reviewer 2 comment No. 1: Thank you for affirming the content and significance of this manuscript.

The epidemiological part would be stronger if the serious disease causing virus types would be mentioned, if this data is available to the writers. For example there are two peaks where the ratio of serious cases is higher compared to other years, 2018 and 2020. 2020 is possibly a coincidence, since the numbers are so small. However, year 2018 is more interesting. In this year incidence was high and the most common cause was CVA6, even compared to other years. Moreover, in that year EV-A71 seems to be very low or completely missing. EV-A71 is known to be a highly pathogenic EV type, causing neurological manifestations which sometimes leads to death.

------Response to reviewer 2 comment No. 2: Thank you for your valuable comments and it has been clarified in the manuscript. Among the 348 severe HFMD cases reported in Shenyang from 2013 to 2023, clinical specimens were collected from 268 cases, of which 232 were positive for enterovirus, with a positivity rate of 86.57%. In 2013, 2015, and 2017, EV-A71 was the most common pathogen in severe HFMD cases, CVA16 was the most common pathogen in severe HFMD cases in 2014 and 2016, and CVA6 was the most common pathogen in severe HFMD cases in 2018 and 2019. After 2020, the number of severe HFMD cases has significantly decreased (Fig. 4C). In addition, CVA2, CVA4, CVA8, CVA10, and CVB2 have all been sporadically detected from severe HFMD cases (Fig. 4D). (Lines 228-235).

Also the question why was CVA16 selected for the genetic characterization is not well addressed. It was not the most common type, CVA6 was. Did this45 have something to do with the pathogenicity? Nothing of this sort was mentioned.

------Response to reviewer 2 comment No. 3: Thank you for your valuable comments, it has been clarified in the manuscript. EV-A71 and CVA16 have always been important pathogens of HFMD. Since 2016, with the launch and application of the EV-A71 inactive vaccine, the prevalence of EV-A71 has shown a decreasing trend, and the pathogen spectrum of HFMD has undergone significant changes, CVA6 has gradually become the main pathogen of HFMD. Since 2013, only one genotype of EV-A71 and CVA6 has been prevalent in China. The C4a sub-genotype of EV-A71 and the D3a sub-genotype of CVA6 are the absolute dominant prevalent genotypes. However, unlike EV-A71 and CVA6, CVA16 has always had two sub-genotypes (B1a and B1b) prevalent in China, which has piqued our curiosity. Therefore, this study also focused on analyzing the evolutionary pattern of CVA16, filling the existing gap in the literature regarding the genetic characteristics of CVA16 in Shenyang City from 2013 to 2023. (Lines 83-92).

The writhers should have payed attention to some details. For example the taxonomical levels are wrong for some parts. The levels of taxonomy are: family: Picornaviridae, genus: Enterovirus, species: Enterovirus A, type or genotype (formerly serotype): Coxsackievirus A16 (and others), genogroup or clade: CVA16 B1, subgenogroup: CVA16 B1a.

------Response to reviewer 2 comment No. 4: Thank you for your valuable comments and corresponding modifications have been made to the levels of taxonomy.

Detailed comments to the manuscript are presented below:

Introduction:

line 45: “various species of Enterovirus (EV)” could it be that here the writers mean type of Enteroviruses, CVA6, CVA16, EV-A71 all belong to species Enterovirus A.

------Response to reviewer 2 comment No. 5: Thank you for your valuable comments and it has been clarified in the manuscript. Hand, foot and mouth disease (HFMD), first reported in New Zealand in 1957, is a common, acute infectious disease caused by various enteroviruses (EV). (Lines 48-49).

line 47: ”mouth herpes”, is strictly speaking caused by the herpes virus. Herpangina or mouth blister would be a better option to be used here.

------Response to reviewer 2 comment No. 6: Thank you for your valuable comments and revised as suggested. The main clinical manifestations are fever accompanied by blisters or rashes on the hands, feet and mouth. (Lines 50-51).

Lines 52-55: This is a very long sentence, consider revising to two or more sentences.

------Response to reviewer 2 comment No. 7: Thank you for your valuable comments, modifications have been made. EV is a non-enveloped, single-stranded RNA virus with a total genome length of approximately 7,400 bp. The genome encodes four structural proteins (VP1-VP4) and seven nonstructural proteins (2A-2C, 3A-3D), and the VP1 region nucleotide sequence is commonly used for genotyping enteroviruses. (Lines 56-60).

lines 56-57:  “four groups, EV-A, EV-B, EV-C, and EV-D” these are species.

------Response to reviewer 2 comment No. 8: Thank you for your valuable comments and revised as suggested. EV infecting humans have been classified into four species: EV-A, EV-B, EV-C, and EV-D. (Lines 60~61).

beginning from line 60: coxsackievirus A16 and enterovirus-A71 are virus types, formerly called serotypes. These can also be called genotypes since the typing is done by sequencing the VP1 like the writers say. Furthermore, the CAV16 can be divided into (at least) three genogroups or clades (subtypes). These again can be divided into subgenogroups or subclades.

------Response to reviewer 2 comment No. 9: Thank you for your valuable comments and revised as suggested. Corresponding modifications have been made to the levels of taxonomy.

Line 61-62: “B1a can be further subdivided into B1a, B1b, and B1c.” The sentence begins with “B1a” – probably should be just “B1”.

------Response to reviewer 2 comment No. 10: Thank you for your valuable comments and revised as suggested. B1 can be further subdivided into B1a, B1b, and B1c. (Line 66).

Materials &methods:

Line 85: herpes is a virus, if by herpes the writers mean a symptom/disease, herpangina or blisters, please change. Herpes can possibly be used to describe a symptom such as blisters, but in scientific text it should not be used.

------Response to reviewer 2 comment No. 11: Thank you for your valuable comments and revised as suggested. Mild cases of HFMD are characterized by markedly elevated temperatures accompanied by a rash or blister of the hands, feet, or mouth. (Lines 97-98). 

Line 105: The sample type pharyngeal swabs, does it include both nasopharyngeal and oropharyngeal swabs? Again, not herpes, rather blisters.

------Response to reviewer 2 comment No. 12: Thank you for your valuable comments and it has been clarified in the manuscript. The types of samples collected included feces, anal swabs, pharyngeal swabs (include both nasopharyngeal and oropharyngeal swab), and blister fluid. (Lines 117-119).

Line 106-107: There is no need to spell out PCR, and furthermore, the test for EV:s is RT-PCR (traditional, real-time, quantitative or not).

------Response to reviewer 2 comment No. 13: Thank you for your valuable comments and revised as suggested. Detection of enterovirus nucleic acid in the above specimens using enterovirus-specific real-time RT-PCR method with nucleic acid detection kits for enteroviruses. (Lines 119-121).

Line 114: It should be said what was extracted form cell culture media. RNA, DNA, nucleic acid or protein, there are more than one option.

------Response to reviewer 2 comment No. 14: Thank you for your valuable comments and it has been clarified in the manuscript. Store the extracted viral RNA at -80°C in a freezer. (Lines 128-129).

Line 116: It is not important in which tubes the RNA was stored, but at what temperature? -20°C or -70/-80°C.  

------Response to reviewer 2 comment No. 15: Thank you for your valuable comments and it has been clarified in the manuscript. Store the extracted viral RNA at -80°C in a freezer. (Lines 128-129).

Results:

The text (lines 156-159) and figure 1 do not correspond to each other. Years 2020-2022 were very low years for HFMD, the increasing trend did not increase throughout 2018 to 2023. This is totally understandable, since it was the time of the COVID-19 pandemic.

------Response to reviewer 2 comment No. 16: Thank you for your valuable comments and it has been clarified in the manuscript. Since 2013, the overall incidence rate of HFMD in Shenyang in Shenyang has fluctuated. From 2014 to 2017, the incidence rate declined year by year. In 2017, the incidence rate reached the lowest value in the past decade; however, in 2018, the incidence rate began to rise to the level of 2015. However, due to the COVID-19 pandemic, the incidence rate declined after 2019, and the incidence rate reaches its highest level since 2013 until 2023. (Fig. 1B). (Lines 168-173).

Figure 1 and most other figures: Figure legend would need a little more details. Also, it should describe what is actually presented in the figures (numbers, percentages etc.). Figure legends with more than one figure in them should first have an overall title. Then after that the description of each subfigure beginning from (A). I have written an example for this in figure 4 below.

------Response to reviewer 2 comment No. 17: Thank you for your valuable comments and it has been clarified in the Figure legends. (Figure 1).

Figure 1: Especially in B and C there is no explanation of what the different bars and lines are. Also, interesting would be in figures B and C, which proportion of cases were detected EV positive.  

------Response to reviewer 2 comment No. 18: Thank you for your valuable comments and it has been clarified in the Figure 1 legend. Time distribution of hand, foot and mouth disease (HFMD) incidence in Shenyang from 2013 to 2023 .(A) Monthly distribution of patients with HFMD in Shenyang between 2013 and 2023. (B) Trends in the annual distribution of patients with HFMD in Shenyang between 2013 and 2023, the dotted line represents the trend line, the solid line represents the incidence rate, and the bar chart is the number of HFMD cases. (C) Yearly distribution of severe HFMD cases in Shenyang between 2013 and 2023, the dotted linerepresents the trend line, the solid line represents the incidence rate, and the bar chart represents the number of severe HFMD cases. (Lines 181-186, figure 1 legend).

Figure 2: to my eye it looks like there are 7 different shades of yellow and brown on the map, but only 5 have the numerical explanation of cases (in figure).  

------Response to reviewer 2 comment No. 19: Thank you for your valuable comments, and we redraw Figure 2.

Line 180: the male-female ratio is in figure 3A, not 3B.

------Response to reviewer 2 comment No. 20: Thank you for your valuable comments and revised as suggested. meaning that the male-to-female ratio was 1.44:1 (Fig. 3A). (Lines 197-198).

Lines 182-183: Why not tell the percentage for the less than 1 year old children of the whole group, like for other age groups. That would make more sense for me.

------Response to reviewer 2 comment No. 21: Thank you for your valuable comments and it has been clarified in the manuscript. The total number of patients aged under 1 year old is 3,187, accounting for only 11.4% of the total number of patients aged under 5 years old. (Lines 200-201).

Figure 3B and in some parts of the text: I don’t know the term “scattered children”, and in the figure named “diaspora” which I possibly understand? Also, there is nowhere that these mean the same, I only concluded this from the numbers that these probably mean the same. If these are the same, please use just one word for this group of children. Maybe this should be better explained, how the scattered children or diaspora children were defined.

------Response to reviewer 2 comment No. 22: Thank you for your valuable comments and it has been clarified. We used scattered children in the manuscript and in the Figure 3.

Table 1: A few points on this table. First, I don’t see the point in the table to present the yearly numbers, the data is already presented in Figure 1. The only striking thing is the high number (and percentage) of severe cases in 2018, but this is also in Figure 1. Other numbers are quite similar and can be discussed in the text and figures. If the writers insist of keeping this table as it is, a summary of all the data should be there. Just add one row to the end of the table with “all together” etc. and add up the numbers. I would move this table to supplementary material or remove it.

------Response to reviewer 2 comment No. 23: Thank you for your valuable comments, Table 1 was removed from the main text as supplementary material.

Table 1 (continued): What has to be corrected or explained are the percentages below the age groups, from what is this the percentage calculated from? For example, year 2020 with very low numbers, percentages of children below 6 years added together is only 0,11%. This is at least not the percentage of the total in year 2020, which is the one that would make sense.

------Response to reviewer 2 comment No. 24: Thank you for your valuable comments. Percentages below the age group are the percentage of children in that age group out of all children in that year.

Line 204: Again, instead of herpes use for example blister fluid or something else.

------Response to reviewer 2 comment No. 25: Thank you for your valuable comments and revised as suggested. A total of 5,531 HFMD samples were received by Shenyang CDC between 2013 and 2023, including blister fluid, anal swabs, and fecal samples. (Lines 219-220).

Figure 4: Be sure the figure legend will be on the same page as the figure itself. The figure legend could be revised, for example, in the following way: First have a title of the picture “The types of enteroviruses detected among HFMD patients in Shenyang, China in years 2013-2023”. Then have the A and B, for example like this “(A) Enterovirus types detected at high rates, (B) less frequently detected types”. I would not use the word pathogens, because this paper deals with enteroviruses in HFMD.

------Response to reviewer 2 comment No. 26: Thank you for your valuable comments and revised as suggested. Types of enteroviruses detected in hand, foot and mouth disease patients in Shenyang, China from 2013 to 2023. (A) Types of enterovirus with a high detection rate. (B) All detected types of enteroviruses. (Lines 239-240; Figure 4 legend).

Figure 4A: why is year 2023 missing? In the discussion is said that the proportion of CVA16 cases has a rising trend, this cannot be seen anywhere. This figure could show some of it, but as I suggested adding a table with the types found might be helpful (possibly as a supplementary).

-----Response to reviewer 2 comment No. 27: Thank you for your valuable comments, We did not collect samples in 2023 due to force majeure. And a table with the number of various enterovirus types detected in hand, foot and mouth disease patients in Shenyang from 2013 to 2022 was added as a supplementary Table 2.

Furthermore, the figure B has all types detected, not only the less commonly detected ones. Also, if I interpret the figures correctly, the next common types are CVA4 and CVA10, which make up about half of the others. That would mean CVA4 would cause about 6-7% of all HFMD enterovirus cases, which is a good proportion of cases, also CVA10 might be 4%?. Should these be in the more common ones? The numbers of virus types found could possibly be presented in a table, which could be as a supplementary. This might interest some of the readers. Possibly also the yearly numbers of virus types in the same table (as a supplementary). 

------Response to reviewer 2 comment No. 28: Thank you for your valuable comments and revised as suggested. A table with the number of various enterovirus types detected in hand, foot and mouth disease patients in Shenyang from 2013 to 2022 was added as a supplementary Table 2.

Figure 5: Details of pictures are very hard to distinguish from each other. Bootstrap values, from the picture it is hard to tell the difference in circle size. Are these circles necessary? The year and genotype colours are too much alike. It is impossible to distinguish several years from each other, for example 2014, 2015 and 2016 as well as 2017 and 2018. It is not even described in figure legend that year colours are on the outer circle and genogroups and sub-genogroups are the inner coloured circle (here again the levels of taxonomy, strain is enterovirus A). These should be used right.

------Response to reviewer 2 comment No. 29: Thank you for your valuable comments and color adjustment has been made to Figure 5. Bootstrap values are indicated by solid circles, and the circle size represents the size of the bootstrap value. The outer circle represents CVA16 from different years, and CVA16 from Shenyang is marked with an asterisk pattern. The inner circle represents different genogroups/sub-genogroups marked with different colors. (Lines 272-275; Figure 5 legend).

Discussion:

These could have been a few sentences about the low Covid-19 pandemic years and the possible shift in the peak of infections. From figure 1 it looks like the shift was seen the first time in 2019, which is pre pandemic. Actually, this peak can be really seen in only two years 2019 and 2023, as seen in fig 1. That cannot really be seen as a trend, only individual years. The peak shift has been mentioned, but nothing on why that might have been.

------Response to reviewer 2 comment No. 30: Thank you for your valuable comments and it has been clarified in the manuscript. The year 2019 before the COVID-19 pandemic was the first time there was a shift towards a peak in the epidemic. In fact, this peak can only be truly seen in two years, 2019 and 2023. During the COVID-19 pandemic, public health measures such as wearing masks, washing hands frequently, and maintaining social distancing have been implemented across the country to prevent and control COVID-19. The school has reduced offline classes, decreased public activities, and implemented travel restrictions, which have reduced opportunities for children and adults to gather together. These measures not only prevent and control COVID-19, but also effectively reduce the spread of infectious diseases such as hand, foot, and mouth disease that are transmitted through close contact. However, with the effective control of the COVID-19 pandemic, strict public health measures previously implemented have gradually been relaxed. In addition, during the COVID-19 pandemic, the spread of HFMD has been restricted due to social isolation and reduced population contact, which may lead to a decrease in the immune level of the population against HFMD related enteroviruses. After the COVID-19 pandemic, the proportion of susceptible populations may increase. All of these may lead to an increase in the incidence of HFMD, resulting in a new peak of incidence in 2023. (Lines 293-309).

Line 259-260: I very much doubt that girls in the ages below 5-6 years would be aware of personal hygiene. At least the European girls are not yet aware at that age, maybe Chinese children are different. It has also been found that boys tend to have slightly more EV and other infections, which is possibly mostly due to immunity.

------Response to reviewer 2 comment No.31: Thank you for your valuable comments and the relevant content without data and reference support has been deleted. The infection rate of boys is 1.44 times that of girls. It is hypothesized that this may be due to differences in susceptibility to host immune status between boys and girls, leading to differences in infection rates between the sexes. (Lines 283-284).

Lines 278-280: The rising trend of CVA16 causing HFMD is not seen anywhere in the data, this is only suggested in the discussion. A table would help with different types causing HFMD would help, and if the rising trend is seen there it could also be in the manuscript itself.

------Response to reviewer 2 comment No. 32: Thank you for your valuable comments and the relevant content without data and reference support has been deleted. The analysis of the pathogens spectrum of HFMD in Shenyang City shows that the CVA16 has been co-prevalent with other enteroviruses in Shenyang City since 2013, accounting for an average of over 10% of the pathogens spectrum of HFMD each year. In recent years, this proportion has exceeded 15%. This indicates that CVA16 is one of the main pathogens of HFMD in Shenyang. CVA16 is also the main pathogen causing severe HFMD cases in Shenyang, especially in 2014 and 2016, it was the first pathogen causing severe HFMD cases in Shenyang. (Lines 317-323).

A table with the number of various enterovirus types detected in hand, foot and mouth disease patients in Shenyang from 2013 to 2022 was added as a supplementary Table 2.

Beginning on line 296: Speculation that changes in dominant genogroup and sub-genogroup to changes in morbidity are interesting. However, it would be helpful to know which virus type, possible genogroup or sub-genogroup have caused the severe cases. If there would be a correlation then it would be more informative and convincing.

------Response to reviewer 2 comment No. 33: Thank you for your valuable comments and it has been clarified in the manuscript. The re-emergence of B1a sub-genogroups is a noteworthy phenomenon, which is caused by the natural process of viral mutation and evolution. The occurrence of this phenomenon may be related to immune evasion, which may mean that the virus has undergone changes, reducing the immune protection previously obtained through natural infection. The re-emergence of B1a sub-genogroups may be more easily transmitted or have increased pathogenicity, which may lead to an increase in the number of cases or the resurgence of the epidemic, thus bringing new prevention and control challenges. (Lines 339-345).

   Line 306: The word summary is mentioned in this sentence twice. Just take away “In summary” from the beginning of the sentence.

Response to reviewer 2 comment No. 34: Thank you for your valuable comments and revised as suggested. Term “In summary” was remove from the beginning of the sentence.